# The transdisciplinary research process and participatory research approaches used in the field of neglected tropical diseases: A scoping review

**Norana Abdul Rahman** [1,2*], **Vaikunthan Rajaratnam**[3], **George L. Burchell**[4], **Karen Morgan**[5], **Mohamed Rusli Abdullah**[6], **Marjolein B. M. Zweekhorst**[1], **Ruth M. H. Peters**[1]

**1** Athena Institute, Faculty of Science, Vrije Universiteit Amsterdam, Amsterdam, The Netherlands, **2** Centre for Research Excellence, Perdana University, Kuala Lumpur, Malaysia, **3** Department of Orthopaedics, Khoo Teck Puat Hospital, Yishun, Singapore, **4** Medical Library, Vrije Universiteit Amsterdam, Amsterdam, The Netherlands, **5** School of Medicine, RCSI-UCD Malaysia Campus, Georgetown, Penang, Malaysia, **6** School of Medical Sciences, University Sains Malaysia, Kubang Kerian, Kelantan, Malaysia

* n.abdulrahman@vu.nl

## Abstract

### Introduction

Neglected tropical diseases (NTDs) comprise a group of twenty diverse diseases or conditions that pose significant public health challenges and adversely impact the quality of life of affected individuals. NTDs are characterised by interconnected biological, social, and environmental factors, which complicate their effective management and eradication. Collaborative research, such as transdisciplinary research (TDR) and participatory approaches that engage scientific, societal, and non-academic stakeholders in co-creating action-driven solutions offer promising strategies to address NTDs. These approaches bridge scientific research with community practices, ensuring evidence-based, contextually relevant interventions. Despite their potential, the application of these approaches in addressing NTDs remains underexplored. This scoping review explores the utilisation of TDR and participatory research approaches to address NTD-related challenges.

### Methods

A systematic search was conducted in PubMed, Web of Science, Embase, and CINAHL, following the JBI methodology for scoping reviews. Data extraction and analysis were performed using JBI SUMARI software, focusing on peer-reviewed published literature reporting the use of TDR and participatory approaches in NTDs, with an emphasis on individual and community perspectives.

### Results

The review examined seventeen articles from Africa, Asia, South America, and Australia, highlighting the increasing use of TDR and participatory approaches to address common

**Data availability statement:** All relevant data are within the manuscript and its supporting information files.

**Funding:** The author(s) received no specific funding for this work.

**Competing interests:** The authors have declared that no competing interests exist.

NTDs such as leprosy, schistosomiasis, rabies, Buruli ulcer, and trypanosomiasis. These approaches engaged diverse stakeholders to develop practical, community-oriented solutions. Key strategies included enhancing public awareness, improving screening programmes, and implementing measures to control NTDs. However, challenges such as fragmented strategies and weak health systems hindered efforts to reduce the burden of NTDs.

## Conclusion

TDR and participatory approaches contribute to a holistic approach in addressing and managing NTD-related challenges by engaging diverse stakeholders and fostering a comprehensive understanding of community needs and on-the-ground realities. The findings demonstrate their effectiveness in translating evidence-informed knowledge into actionable interventions to benefit affected individuals and their communities.

### Author summary

Neglected tropical diseases (NTDs) are a group of twenty complex diseases and conditions that disproportionately affect marginalised populations, causing severe health, social, and economic hardships. These diseases are characterised by interconnected biological, social, and environmental factors, often involving animal reservoirs, which complicates their prevention and control programmes. Addressing these challenges requires innovative and inclusive approaches. This scoping review explores the potential of transdisciplinary research (TDR) and participatory approaches in addressing the multifaceted challenges of NTDs. TDR brings together knowledge from multiple disciplines, while participatory methods actively involve affected individuals and communities in the co-creation of solutions. Together, these approaches encourage collaboration, improve public awareness, and address the broader social and environmental contexts influencing NTDs. By improving screening programmes and implementing effective control measures, TDR and participatory approaches can contribute to addressing the complexities of NTDs and reduce their burden. This review emphasises the value of involving diverse perspectives to create sustainable, community-centred interventions that improve the well-being of affected populations and provide actionable insights for researchers, practitioners and policymakers.

## Introduction

The World Health Organisation's (WHO) list of NTDs comprises twenty diseases and conditions caused by parasites, viruses, bacteria, and toxins, including snake bites [1]. Soil-transmitted helminthiases, schistosomiasis, and food-borne trematodiases are the most prevalent NTDs globally [2], while visceral leishmaniasis, rabies, and Chagas disease are the leading causes of NTD-related deaths [3]. These diseases display diverse clinical presentations and transmission patterns [1,4] influenced by various factors such as politics, economics, culture, society, and the environment, including climate change [5–7].

NTDs disproportionately affect populations in resource-limited settings [1], impacting more than 1.7 billion people [8] in tropical and subtropical regions of Africa, Asia, and South

America [9,10]. These countries bear the burden of multiple NTDs simultaneously, with individuals often being infected with multiple diseases concurrently [8]. This perpetuates cycles of poverty [11,12], contributing to malnutrition, cognitive impairment, and long-term disabilities, impacting both children and adults [13–15]. Moreover, the chronic and slow-progressing nature of NTDs contributes to social isolation, mental health issues, and diminished quality of life [8,9,16]. Additionally, the stigma associated with NTDs poses a constraint to accessing care and support [11,12].

Although NTDs are preventable, they are challenging to manage because they are related to the environment, and many of the pathogens have a zoonotic component to their complex lifecycles. Animal reservoirs in NTDs are a persistent source of ongoing transmission and re-infection [10]. NTDs thus continue to pose significant challenges to public health. The substantial progress in combatting NTDs, achieved through expanded global initiatives, enhanced collaboration, and innovative research [11,15], has improved disease surveillance, treatment, and prevention [17], benefitting over one billion people through mass treatment interventions from 2016 to 2019 [18]. Forty-seven countries have eliminated at least one NTD as of December 2022, with more countries striving to reach this target [18].

However, significant work remains to address the unmet needs, priorities, and challenges in this ongoing battle [17,19]. They include limited access to treatment and healthcare services for millions of affected individuals due to geographical, socio-economic, and infrastructural barriers [5,20]. These services often lack interdisciplinary coordination, leading to inefficiencies in integrating NTD programmes with other healthcare systems and initiatives [7,17]. Weak health systems and the complexity of targeting animals and the environment pose additional challenges [21,22]. Addressing these unmet needs requires sustained political commitment, adequate funding, and a comprehensive approach that considers the social, economic, and environmental factors influencing NTDs. By continuing to build on achievements and addressing ongoing challenges, it seems possible to make further progress in controlling and managing NTDs globally.

The adoption of TDR and participatory approaches, which involve diverse stakeholders and account for local contexts, provides a comprehensive understanding of the complex problems related to NTDs [20]. Participatory research emphasises collaboration by actively involving affected individuals and communities to shape interventions that address their specific needs and circumstances [23,24]. TDR complements this approach by transcending traditional disciplinary boundaries, integrating knowledge from diverse fields, and fostering collaboration across sectors, including at the local level. Through mutual learning and co-creating solutions with stakeholders, TDR ensures that interventions are contextually relevant, holistic, and sustainable. Participatory research often serves as a foundational component within TDR projects, offering community insights that inform interdisciplinary strategies. Frameworks like One Health, a multisectoral approach [25,26], exemplify TDR by recognising the interconnectedness of humans, animals, and the environment, enabling the development of sustainable solutions informed by diverse perspectives [25].

Despite these strengths, collaboration with multiple stakeholders can be challenging due to differing knowledge or epistemologies, power imbalances, and resource constraints, such as time, finances, and limited multidisciplinary expertise [27,28]. However, TDR and participatory approaches promote knowledge sharing among scientific, societal, and non-academic stakeholders [29,30]. They facilitate capacity building and the efficient translation of research findings into action-driven solutions [17,31], optimising resource utilisation that aligns with community-specific needs and encourages community participation in combatting NTDs [32,33].

To address the complex challenges associated with NTDs, TDR, including frameworks like One Health and participatory approaches, offers valuable pathways. Hence, to fully harness their potential, we need to better understand their respective values, challenges, and interplay when applied to NTDs [34]. However, the extent of their application and impact on NTD interventions in the literature remains unclear. Given the interdisciplinary and multi-sectoral nature of these approaches, a scoping review is appropriate to explore their implementation and effectiveness [35]. This scoping review explores the utilisation of TDR and participatory research approaches to address NTD-related challenges. Specifically, it examines the contexts in which these approaches have been applied, the nature of stakeholder collaboration, and the factors influencing their implementation. It also identifies key facilitators, barriers, and the effectiveness of these approaches in addressing the complex issues associated with NTDs. The research questions guiding this review are detailed in the Methods section.

## Working definitions

Essential to this study is the establishment of operational definitions for standard terms (S1 Appendix), facilitating consistent understanding among readers and researchers. These definitions not only improve communication, collaboration, and literature integration but also ensure accurate interpretation of findings, thereby strengthening the study's quality and rigour.

Due to the lack of a universal definition for TDR and the varying perspectives on methodological standards [35–37], this review considered a study as TDR if it demonstrated a shared goal of addressing complex real-world problems through the integration of diverse knowledge systems and collaboration between scientific, societal, and non-academic stakeholders [30]. To qualify as TDR, the study must also include individuals or their communities directly impacted by NTDs, ensuring a systemic approach to co-creating holistic and sustainable solutions. TDR, including One Health, and participatory approaches are inherently interconnected and will be considered collectively in this study. These approaches are not mutually exclusive; TDR, including One Health, often integrates participatory elements by engaging affected individuals, communities, and stakeholders to develop solutions grounded in real-world contexts. Similarly, participatory approaches can leverage the integrative and systemic perspective of TDR to expand their scope beyond localised challenges. Together, they form a complementary framework for addressing the complex and multi-faceted issues associated with NTDs.

## Methods

This scoping review protocol was registered with the Open Science Framework (DOI 10.17605/OSF.IO/WQUS4). The review followed the Johanna Briggs Institute (JBI) methodology for scoping reviews [38,39], ensuring a systematic approach to identifying and synthesising relevant literature. Data extraction and analysis were conducted using the JBI SUMARI software. [40]

The following review questions were identified to guide the search strategy and ensure all types of studies and a broad range of literature were captured:

1. In what contexts were the TDR and participatory approaches used to identify and prioritise the complex problems faced by individuals with NTDs?

2. In the NTD studies, how were the TDR and participatory approaches utilised to address the division of social domains, including the stakeholders involved, the collaboration between scientists, societal and non-academic actors, and their power dynamics and the strategies to address them?

3. What were the facilitators for using the TDR and participatory approaches to address the problems associated with NTDs?

4. What were the barriers to implementing TDR and participatory approaches in addressing the problems associated with NTDs?

5. How effective or adequate were the TDR and participatory approaches in addressing the problems associated with NTDs?

## Search strategy

The search strategy was carried out by GLB and NAR to identify relevant peer-reviewed publications for inclusion in the scoping review. A systematic search was conducted in the databases: PubMed, Embase.com, Clarivate Analytics/Web of Science Core Collection, and Cumulative Index to Nursing and Allied Health Literature (CINAHL) (30-09-2021). (S2 Appendix) The search strategy included all identified keywords and free text terms for (synonyms of) 'transdisciplinary research' and (individual names of diseases and synonyms of) ' Neglected Tropical Diseases'. These keywords were adapted for each included information source. The search strategy was also performed with the descriptors from the international vocabulary used in the health area, MeSH - Medical Subject Headings created by the National Library of Medicine for literature indexed on MEDLINE, combined through Boolean operators. Pure animal studies were excluded from the search. The search strategy can be found in the supplementary information (S2 Appendix). Duplicate articles were excluded using the R-package "ASYSD," an automated deduplication tool [41], followed by manual deduplication in Endnote (X20.0.3) by the medical information specialist (GLB).

The PCC framework (Participants, Concept, and Context), as recommended by JBI, as shown in Table 1, was used to qualify the eligibility criteria. [39]

Table 1.  The PCC Framework (Eligibility Criteria).

| Participants | **This scoping review considered all peer-reviewed published articles that specifically incorporated the perspectives of individuals or their communities directly impacted by NTDs and any other participants directly or indirectly affected or involved with NTDs, regardless of geographical location. The review did not include purely animal or environmental studies.** |
|---|---|
| Concept | This review considered all peer-reviewed published articles that reported on, referred to, or explained the TDR and participatory approaches involving multiple stakeholders, specifically focused on affected individuals and their communities, to study or address problems associated with NTDs. These should be mentioned in the titles and abstracts, not just confined to the main text. |
| Context | This review considered all published articles in any setting, from any geographical location, that focused on using the TDR and participatory approaches to manage problems associated with NTDs. These approaches should be referred to or explained in the titles or abstracts of the articles. |
| Type of Studies | There were no restrictions on the type of research or timeframe for the included studies as long as they focused on TDR and participatory approaches in NTDs, specifically involving individuals directly affected by NTDs.<br>The review considered articles published in English. Due to resource limitations for translation, only articles with an existing English translation from another language were included. |

## Study selection

Following the search, all identified citations were uploaded into EndNote 20 (Clarivate Analytics, PA, USA) citation management platform, and all duplicates were removed. This review used a two-stage screening process to assess the relevance of studies identified in the search. Only the title and abstract of citations were reviewed at the first level of screening. A trial run was first conducted, following which two reviewers (NAR and RV) independently screened the titles and abstracts for assessment against the inclusion criteria for the review. Due to resource constraints, the only deviation from the JBI protocol was the decision not to search for unpublished literature and not to carry out a hand search of the reference lists for additional studies.

All potentially relevant articles were subsequently retrieved in full. Their citation details were imported into the JBI SUMARI software (JBI, Adelaide, Australia). [42] The two reviewers (NAR and RV) further assessed the full text of the included studies in detail. Any disagreements were resolved through discussions between the two reviewers. Seventeen studies [42–58] were included in this review and are listed in S3 Appendix. Critical appraisal or risk of bias assessment was not conducted in this scoping review because the aim was to map the available evidence on the use of the TDR and participatory approaches to address the complex problems associated with NTDs, not the quality of the included articles.

## Data Extraction, analysis, and presentation table

The two reviewers conducted the trial run of the draft extraction sheet to ensure relevant data were extracted. The instrument was fine-tuned and accepted for the data charting process. The data extracted provided a logical and descriptive summary of the results relevant to using the TDR and participatory approaches in NTDs. Data were extracted from the included studies by NAR using the refined data extraction tool (S4 Appendix) and were checked by RV for accuracy. During the data extraction, disagreements between the reviewers were resolved through discussions to reach an agreement. Authors of papers were also contacted to request for missing or additional data. The data extraction table (S5 Appendix) summarised the characteristics of the included studies. It provided a concise overview of the literature.

## Results

### Study inclusion

The electronic database search identified 3071 records, of which 1769 duplicate articles were excluded. This review considered peer-reviewed published articles that reported on, referred to, or explained the TDR and participatory approaches involving multiple stakeholders (scientific, societal, and non-academic), specifically individuals or their communities who are directly impacted by NTDs, to study or address challenges associated with NTDs in their titles or abstracts. After screening the titles and abstracts of the remaining 1302 articles, 1249 were excluded for failing to meet the selection criteria. Subsequently, fifty-three articles were included for full-text screening, but one article was excluded since it was not a peer-reviewed article. Out of the fifty-two articles, in total, thirty-five were removed, consisting of one duplicate and one article in Portuguese without an English translation. Seven articles did not meet the inclusion criteria for phenomena of interest, as their focus was not on exploring TDR and participatory approaches to address the challenges associated with NTDs. Another twenty-six did not meet the study design criteria, as they did not include individuals, or their communities directly impacted by NTDs. This resulted in seventeen articles being included in the review (S3 Appendix). The search and study selection process was presented in a Preferred Reporting Items for Systematic Reviews and Meta-Analyses Extension for Scoping Reviews (PRISMA-ScR) flow diagram (Fig 1) [59].

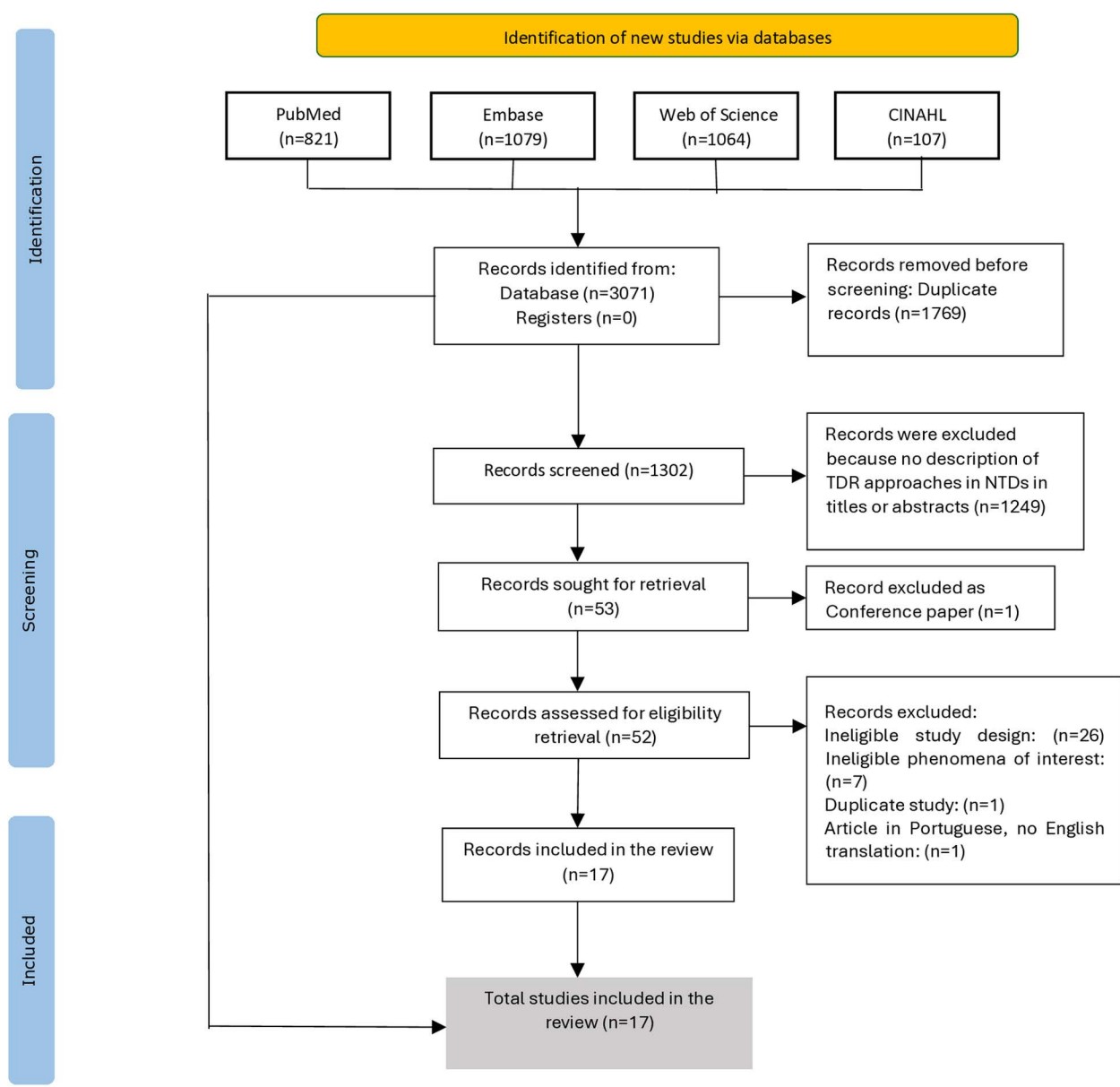

**Fig 1. PRISMA-ScR flow diagram of search and study selection process.**

## Characteristics of the included studies

S5 Appendix presents the characteristics of the seventeen studies [42–58] included in this review, highlighting the application of TDR and participatory approaches in addressing the challenges associated with NTDs. Each study reported the number of participants involved in the research. The participant composition across these studies included scientists, experts, and academicians, as well as government representatives, international and local institutions, non-governmental organisations (NGOs), the private sector, health workers, community leaders, affected individuals, and community members. The diversity emphasised the participatory and inclusive nature of the research approaches. The data collection methods varied across the studies and included in-depth interviews, focus-group discussions, surveys,

activities, and review of medical records. Some studies employed mixed methods to capture diverse perspectives and ensure a comprehensive understanding of the research context.

The articles included in this review spanned nearly three decades. Fig 2 demonstrates the increasing traction of the use of TDR and participatory approaches, particularly in recent years. With the exception of the article by El Katsha et al. [48], published in 1993, the majority were published after 2006. Ten of the 17 articles were published between 2018 and 2021 [43–46,49,51,53,54,57,58], and six between 2006 and 2016. [43,49,51,53,56,57]. This trend emphasises the growing interest in using these approaches to address the multifaceted challenges of NTDs.

## TDR and participatory approaches

All the studies included in this review [42–58] have featured TDR and participatory research approaches as their central component, yet a precise definition for TDR was absent. Notably, only Peters et al. [56] used the definition of TDR as provided by Klein in 2001 [30]: *Transdisciplinarity has been defined as "a new form of learning and problem-solving involving cooperation between different parts of society and science in order to meet complex challenges of society. Transdisciplinary research starts from tangible, real-world problems. Solutions are devised in collaboration with multiple stakeholders."*

The research process in all the studies [42–58] commenced with a needs assessment from the community perspective. They indicated that, as a fundamental step, researchers prioritised understanding and addressing the specific requirements, challenges, and perspectives of the communities directly affected by NTDs. The studies highlighted the commitment of the

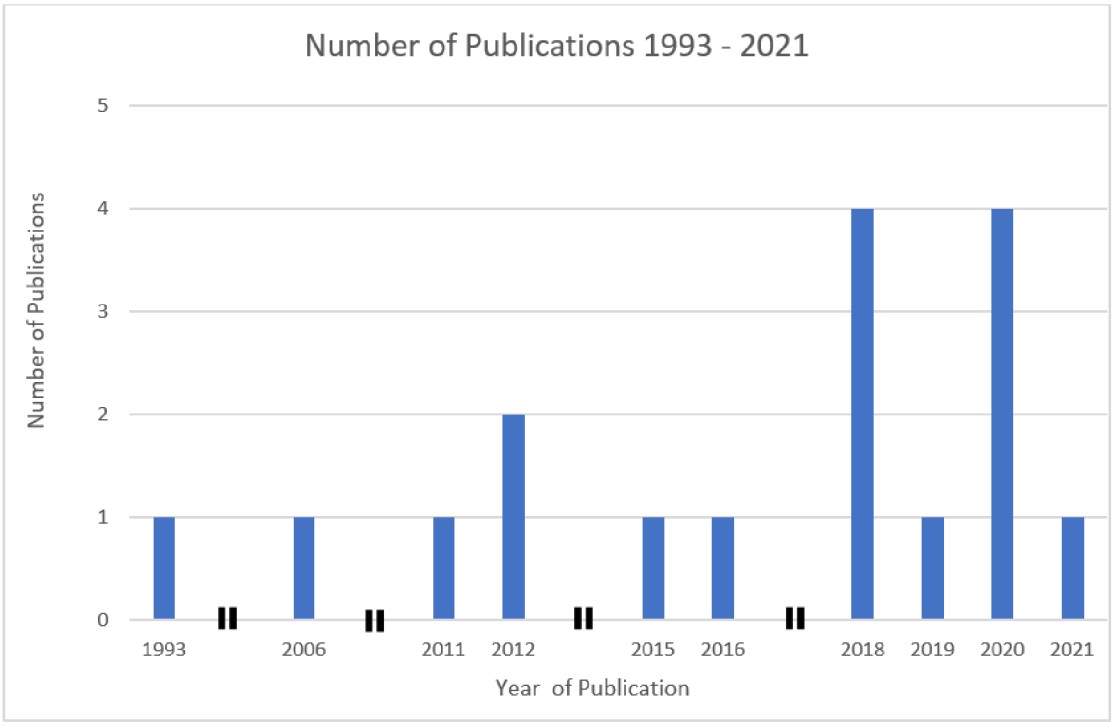

Key: ▮▮ Means years with no data

**Fig 2. The number of peer-reviewed publications on TDR and participatory approaches in NTDs from 1993 to 2021.**

researchers to ensure that the research objectives and interventions aligned closely with the actual needs and experiences of the community. For example, Degeling et al. [47] engaged with communities facing the highest risk in the event of a rabies incursion, emphasising the significance of actively listening to individual and community concerns. They stressed the need to address educational and infrastructural requirements to empower these communities to respond effectively. This approach was essential for them to design effective, contextually relevant, and culturally sensitive interventions, as the researchers placed the community at the forefront of their research process.

The review showed that all the studies approached their problem-solving process using the three distinct phases (or Phase Model Concepts of TDR) introduced by Pohl and Hirsch Hadorn. [60] These phases involved collaborative problem identification and structuring, investigation and co-creation of knowledge/solutions, and integration and application, focusing on implementing the co-created results into practice. Peters et al. [56] used the Interactive Learning and Action (ILA) method [61], which aligned with the conceptual framework but with some variations.

Monitoring and evaluating (M&E) interventions were crucial for assessing their impact and effectiveness, as they offered insights into whether the interventions achieved their intended outcomes and helped identify areas for improvement or modification. However, only nine studies mentioned their future directions [42–44,47,48,53–55,58]. These studies employed TDR and participatory approaches to integrate M&E frameworks, ensuring that the interventions were not only responsive to community needs but also continuously assessed for effectiveness and adaptation. The work of Awah et al. [45] demonstrated the importance of comprehensive evaluation of interventions. The study showed that continuous M&E ensured accountability, enabled evidence-based decision-making and contributed to the ongoing refinement of intervention strategies, ultimately maximising their positive effects on the targeted individuals or communities. The TDR and participatory approaches in these studies focused on addressing the challenges presented by NTDs and the related social determinants. These studies chose the TDR and participatory approaches for their potential to offer effective problem-solving and develop sustainable, community-centric solutions that were relevant and aligned with the needs of affected communities.

### This scoping review aimed to answer the review questions as follows

1. Contexts in which TDR and participatory approaches were used to identify and prioritise the complex problems faced by individuals with NTDs

The studies included in the review showed the diverse contexts in which TDR and participatory approaches were implemented to address the multifaceted challenges associated with NTDs. These contexts span geographic, socioeconomic, and thematic dimensions, highlighting the adaptability and effectiveness of these approaches. The geographical scope of the studies was diverse: they were conducted across Africa, Asia, Australia, and South America, covering a range of socioeconomic contexts, as outlined in Table 2. Most countries were classified as low- and middle-income nations, except for Australia. Specifically, nine studies were conducted in Africa and five in Asia; one took place in Australia, and two studies adopted a multi-country approach.

This review comprised fifteen studies conducted in rural areas [42–48,50,52–58], six of which also encompassed urban settings [43,46,47,51,55,56]. Only two studies focused solely on urban areas. both conducted in Asia [50,52]. This distribution reflects the applicability of TDR and participatory approaches across diverse geographic and socioeconomic environments.

**Table 2. Country/Countries in which the research was conducted in the reviewed articles.**

| Countries | No of Articles | Citations |
| --- | --- | --- |
| Africa | 9 | [42,44,47,48,52,53,56–58] |
| Asia | 5 | [43,49–51,55] |
| Australia | 1 | [47] |
| Multiple countries:<br>-Mozambique, Nepal and Peru<br>-Nigeria and Liberia | 2 | [46,55] |

Table 3 provides an overview of the NTDs addressed in the studies. Leprosy was the most studied, appearing in four articles. Schistosomiasis and general NTDs were each addressed in three articles. There were two articles each on Buruli ulcer, rabies, and trypanosomiasis and one on trachoma. These articles explored a variety of issues such as surveillance strategies, case-finding, treatment methods, access to multi-drug therapy, integration of services with primary care, initiatives that promoted early diagnosis and treatment, and the empowerment of individuals affected by leprosy to reduce stigma.

All the studies [42–58], emphasised collaboration among diverse stakeholders from various disciplines. These approaches aimed to achieve comprehensive problem-solving and effective intervention strategies, particularly focusing on addressing social determinants of health and health system challenges. Key initiatives included improving access to treatment, enhancing surveillance, and reducing stigma through community engagement. They were used within the context of health systems strengthening, which created an enabling environment for sustainable solutions by improving health practices, policies, and infrastructure. Several articles in this review [46,55,58] specifically addressed the need for robust health systems to support effective and long-term interventions for NTDs.

Exploring the implementation of TDR and participatory approaches across diverse contexts revealed the various factors influencing their effectiveness as research methodologies. Despite these contextual variations, a shared objective among all studies was the enhancement of disease control programmes and the implementation of sustainable prevention initiatives. Central to these efforts were community engagement and the integration of local skills and knowledge [42–58]. For example, El-Katsha et al. [48] conducted a study in a rural area focusing on Schistosomiasis control, highlighting the importance of holding meetings between researchers, staff members, and villagers. These interactions fostered a deeper understanding of persistent exposure to contaminated canal water and explored strategies to minimise such

**Table 3. NTDs covered in the review.**

| Diseases | No of Articles | Citations |
| --- | --- | --- |
| Leprosy | 4 | [44,51,52,56] |
| Schistosomiasis | 3 | [48,49,54] |
| Buruli ulcer | 2 | [43,45] |
| Rabies | 2 | [47,50] |
| Trypanosomiasis | 2 | [57,62] |
| Trachoma | 1 | [58] |
| NTDs (general, not specified) | 3 | [46,53,55] |

contact. This context demonstrated the active involvement of affected individuals or their communities alongside the scientists, practitioners, and advocates, highlighting the collaborative nature of TDR and participatory approaches and its reliance on inclusive participation.

2.  Application of TDR and participatory approaches to address the division of social domains, including the stakeholders involved, the collaboration between scientists, societal and non-academic actors, and their power dynamics and the strategies to address them

All the studies in the review implemented the TDR and participatory approaches to address the division of social domains involving diverse stakeholders, collaboration dynamics, and power structures. The studies began by identifying and engaging various stakeholders, documenting their composition and the number of participants in the research to ensure a diverse representation for understanding the multifaceted challenges of NTDs. According to the researchers, engaging these stakeholders from the outset helped ensure that research questions aligned with the goals of the project and facilitated a holistic understanding of NTDs. The studies indicated that collaboration between stakeholders was crucial in fostering mutual learning, facilitating the co-creation of innovative solutions, and developing effective strategies aligned with the specific research goals and objectives to address the complexity of NTD-related challenges. Table 4 shows the diverse stakeholders who contributed their scientific and societal knowledge, values, and expertise to generate valuable insights into various NTDs.
While community engagement in disease control is widely recognised for mitigating the impact of NTDs on affected individuals, all 17 studies [42–58] involved the community as a stakeholder from the outset. The review showed that the rationale for including the community was its effectiveness in building trust, raising awareness, addressing barriers, enhancing programme outcomes, and ensuring cultural relevance. [63]

The review revealed that power dynamics, inherent in TDR and participatory approaches due to collaborations with diverse stakeholders, can create hierarchical structures, influence motivations among participants, and affect decision-making. These dynamics must be addressed to foster an equitable research environment. However, only three studies [53–55] in the review highlighted and discussed the challenges these power dynamics present. Onasanya et al. [54] conducted stakeholder mapping to understand how influential key players shaped power dynamics during the collaborative development of a diagnostic device. Similarly, Ozano et al. [55] emphasised the importance of handling power dynamics and fostering participation to establish equitable and sustainable partnerships. Peters et al. [56] explored, identified, and addressed the different mindsets of stakeholders to reduce stigma in leprosy-affected individuals. They demonstrated the importance of upholding fundamental principles to establish a stronger foundation for their work. These three studies [53–55] highlighted the importance of recognising and navigating power dynamics and perceived it as central to the success of their projects, while the remaining studies did not refer to them explicitly.

All the studies, including those that did not refer to power dynamics, adopted various strategies to reach more equitable power distributions. These included efforts to enhance the capacities of the local communities through education and training programmes, the creation of platforms for dialogue and negotiations, and transparent decision-making processes. TDR and participatory approaches also encouraged reflexivity among researchers to examine their positions of power and consider how these might have impacted the collaborative efforts. Moreover, the collaboration among scientists, societal and non-academic actors was crucial in ensuring that the social aspects of NTDs were considered and integrated into the broader public health agendas. By actively engaging diverse stakeholders throughout the research process,

**Table 4.  Stakeholders included in the reviewed articles.**

| Stakeholders | No of articles | Citations |
|---|---|---|
| Public/community, including patients and ex-patients/Patient support groups | 17 | [42–58] |
| Health Officials | 17 | [42–58] |
| Government Officials | 13 | [42,44,45,47,49–56,58] |
| Researchers, Academicians, Sociologists | 8 | [43,44,48,53–56,58] |
| Non-Governmental Organisations (NGOs) | 8 | [44,49–51,53–55,57] |
| Veterinarians, Animal Health, Experts in wildlife, Animal Husbandry | 5 | [47,48,50,57,62] |
| Teachers | 5 | [46–48,53,55] |
| Environmental experts | 4 | [43,48,57,62] |
| Private Sector/Development partners | 4 | [50,53,55,57] |
| Banks/donors | 4 | [53–56] |
| Traditional healers | 3 | [45,54,57] |
| International experts/International organisations | 2 | [49,51] |
| Religious leaders | 2 | [54,56] |
| Pharmaceutical/drug donors | 1 | [51] |

the studies tried to promote relevance, maximise impact, and establish fair partnerships that drove substantial and meaningful change.

3. Facilitators for using TDR and participatory approaches to address the challenges associated with NTDs

The studies in this review [42–58] provided valuable insights into the facilitators of using TDR and participatory approaches to address the multifaceted challenges of NTDs. Strong collaborations among stakeholders, particularly the affected individuals and their local communities, were perceived to foster a holistic understanding of NTDs. This interdisciplinary and community collaboration facilitated the development of shared goals and visions for addressing NTD-related issues. [42–58] Engaging the community in disease control was deemed crucial for reducing the impact of NTDs on affected individuals and ensuring that interventions were culturally sensitive, relevant, and effective [42–58]. Ackumey et al. [43] demonstrated that collaboration and networking among stakeholders, including patients and carers, and the training of health staff and other stakeholders improved the delivery of health programmes. Freudenthal et al. [49] assessed the sustainability of disease interventions within the community, highlighting the importance of community involvement, including the younger population, in disease control and prevention initiatives. These approaches, as highlighted by the two studies, recognised the significance of local knowledge and empowered communities in the fight against NTDs.

A recurring challenge in TDR and participatory approaches was managing power dynamics. The studies [53–55] addressing this issue demonstrated their positive impact on equitable collaborations, decision-making, and project success. Managing these dynamics was perceived as central to fostering effective and sustainable partnerships.

Partnerships were another facilitator, enabling diverse stakeholders to coordinate efforts in strengthening disease elimination plans and ensuring the long-term sustainability of NTD

projects. [43,45,48,49,55,57] By bringing together expertise from various disciplines and sectors, these partnerships facilitated the development of comprehensive strategies and efficient resource allocation for NTD control. The review also demonstrated the interconnectedness of human, animal, and environmental health. Integrating these factors improved access to healthcare for affected individuals, enhanced animal welfare, and contributed to more comprehensive and effective strategies in NTD prevention and control. [43,47,48,57,62]

The availability and accessibility of WHO-recommended antibiotics and vaccines through government agencies and private organisations also played a crucial role in facilitating TDR and participatory approaches to address challenges associated with NTDs. These resources provided standardised and effective treatment options for diseases such as leprosy, Buruli ulcer, schistosomiasis, trypanosomiasis, and rabies [42–44,47–50,56,58]. This availability offered researchers and practitioners reliable treatment options for the diseases under investigation. Researchers collaborated with local stakeholders to align interventions with the needs of affected populations, as shown by Apte et al. [44] where single-dose rifampicin, as post-exposure prophylaxis, improved community morale through ease of administration and enhanced training and supervision.

The flexibility and adaptability of the TDR and participatory approaches across diverse contexts and research settings facilitated their use in addressing NTD challenges. Some studies [42,44,47,48,53–55,58] showed how this flexibility permitted the incorporation of tailored strategies to address unique challenges presented by various diseases in different environments. Awah et al. [45] showed how lessons from a proof of concept study informed modifications in a larger pilot study to enhance community engagement, improve communication, and solve problems at the grassroots level. Similarly, Freudenthal et al. [49] demonstrated the adaptability of participatory approaches in initiatives involving teachers as co-researchers in schistosomiasis education, connecting health and education sectors and empowering stakeholders to achieve meaningful, sustainable outcomes. These examples showed that the effective use of TDR and participatory approaches in addressing NTDs was facilitated by interdisciplinary and community collaboration, managing power dynamics, strategic partnerships, resource availability, and adaptability. These elements contributed to enhancing the relevance and success of the research efforts.

4. Barriers to using the TDR and participatory approaches to address the problems associated with NTDs

Some studies in the review identified several barriers that hindered the effective use of TDR and participatory approaches in NTD-associated problems. [42,44–48,50–58] These barriers disrupted key processes essential to the collaborative approaches, such as fostering stakeholder collaboration, integrating diverse perspectives, and ensuring equitable decision-making. By impeding these processes, these barriers limited the ability of TDR and participatory approaches to address the multifaceted challenges of NTDs effectively.

A significant barrier was the lack of clear guidelines and understanding for conducting research using TDR and participatory approaches, which hindered the adoption of standardised methods, resulting in inconsistencies in research practices. For example, Onasanya et al. [54] highlighted that a lack of guidelines and tailored frameworks for stakeholder analysis in NTD research and the co-creation of diagnostic devices compelled them to rely on their understanding of the healthcare system and disease context in the country, supplemented by contextual inquiry. Limited stakeholder understanding of how TDR and participatory approaches work resulted in a lack of shared problem-framing, as differing perspectives on the same issue led to fragmented approaches. Kuipers et al. [52] identified delays in disease diagnosis caused by variations in the stakeholders' experiences, roles, and interpretations of the problems, contributing to disparate solutions.

Power imbalances among stakeholders were another barrier that restricted meaningful engagement, equitable representation, and effective decision-making. Ensuring fair stakeholder representation, addressing unhealthy competition, and managing differing perspectives among stakeholders were essential to overcoming these challenges [53–55]. Peters et al. [56] emphasised the importance of promoting awareness and efforts to achieve equity within the project team through actions like inviting affected individuals to meetings, creating opportunities for sharing stories, and encouraging questions to promote inclusive decision-making.

Other barriers to the use of TDR and participatory approaches in NTDs were the lack of disease burden data and a weak surveillance system, which hindered effective communication and information sharing among stakeholders, which are central to collaborative efforts [45–54,56,58]. Reid et al. [57] highlighted how insufficient knowledge about treatment options and the socio-economic impacts of NTDs on families led stakeholders to underestimate the disease burden, resulting in a lower prioritisation for funding or public health interventions. Fragmented efforts to integrate knowledge across different fields, such as animal and environmental health in rural areas, and the failure to recognise targeting animals as an alternative approach to reducing disease burden in humans compounded these issues [46–49,56,58]. Gautam et al. [50] highlighted these fragmented efforts in rabies control, noting that uncoordinated stakeholder actions further complicated effective disease management.

Financial and resource constraints posed another barrier, particularly in low-income countries. Limited funding for research, reduced political commitment, poor infrastructure, and a lack of human resources and trained personnel further hampered the ability to conduct comprehensive and impactful TDR and participatory studies [42,45–47,50,51,53,58] Degeling et al. [47] emphasised the significance of addressing the inadequate infrastructure in indigenous communities to better prepare at-risk populations for potential rabies incursion.

Other barriers included cultural perceptions, mistrust in the healthcare systems, and reliance on traditional healers, which undermined stakeholder engagement, which is central to the use of TDR and participatory approaches in NTDs [44,46–50,52–59,62]. Ackumey et al. [43] highlighted concerns such as fear of amputations, loss of livelihood, and additional costs associated with hospitalisation, which compounded difficulties in ensuring meaningful stakeholder participation and equitable representation in the study.

The absence of systematic monitoring and evaluation (M&E) to assess the impact and effectiveness of interventions in NTD-related challenges posed a significant barrier to the effective application of TDR and participatory approaches. Without structured assessments, it was difficult to understand the success or shortcomings of interventions, thereby limiting the ability to make informed decisions about future improvements or modifications. Only nine studies [44–46,49,50,55–57,59] within this review reported M&E of intervention impacts. emphasising the need for more robust evaluation frameworks.

The listed barriers collectively highlighted the challenges faced in utilising TDR and participatory approaches to effectively address NTDs. Addressing these barriers requires standardised guidelines, improving communication and planning, developing diagnostic capabilities and healthcare access, ensuring adequate surveillance, allocating sufficient resources and time, fostering cultural sensitivity, and ensuring equitable stakeholder representation to ensure the effectiveness of these approaches.

5. Effectiveness of the TDR and participatory approaches in addressing the problems associated with NTDs

The TDR and participatory approaches employed in the studies reviewed [44–59,62] demonstrated their potential to address complex challenges and improve outcomes in the fight against NTDs. These approaches generated new knowledge that contributed to positive

changes in the control and management of NTDs. By tailoring interventions to the specific needs and challenges of the communities, TDR and participatory approaches proved effective. Their flexibility and adaptability allowed meaningful engagement with diverse stakeholders, enabling them to address the problems associated with NTDs effectively.

The alignment of interventions with the local community boosted acceptance, fostered participation, and enhanced engagement among affected populations. Awah et al. [45] successfully established a Community of Practice for disease control, which improved collaboration between community members and clinic staff, resulting in increased referrals and sustained stakeholder engagement. Similarly, El Katsha et al. [48] demonstrated how involving the community can effectively integrate local knowledge and resources into intervention strategies, strengthening disease control initiatives. Jaeggi et al. [51] further emphasised the importance of sustaining an integrated approach to leprosy services within the primary care framework, highlighting how this would help ensure long-term effectiveness.

Flexibility and cultural sensitivity further enhanced the success of TDR and participatory approaches. Freudenthal et al. [49] adapted their initial focus by involving teachers as research partners rather than project assistants. This flexibility to modify their research led to a new schistosomiasis education curriculum and stronger school-community ties. It demonstrated the ability to adjust strategies in response to evolving circumstances, which could contribute to sustainable outcomes. While TDR and participatory approaches effectively addressed the challenges of NTDs, many studies lacked follow-up plans, which are critical for assessing impacts to ensure their effectiveness. This highlights the importance of iterative evaluation cycles to refine and enhance TDR and participatory initiatives over time.

## Discussion

In this scoping review, we explored the utilisation of TDR and participatory research approaches to address NTD-related challenges. By examining 17 studies from 14 countries across Africa, Asia, South America, and Australia, the review sought to understand how these approaches were implemented and their effectiveness in creating meaningful and sustainable societal transformations. [44–59,62] Through this analysis, our key findings included the diverse contexts in which these approaches were applied, the facilitators and barriers encountered, and their effectiveness in producing community-centric solutions. While TDR and participatory approaches were central to the studies, their roles and contributions varied, highlighting the methodological and conceptual flexibility required to address complex health issues.

Despite its growing application, TDR lacks a universal theory, methodology or definition. [64] The absence of standard definitions and frameworks for TDR poses a significant barrier, which poses challenges for consistent implementation. Additionally, the lack of a shared understanding among stakeholders often hinders effective collaboration, resulting in differences of opinion, miscommunication and fragmented approaches in addressing NTD challenges. As Haring et al. [65], highlight, differing objectives and perspectives among partners often create tensions, affecting the course and outcomes of collaborative healthcare projects. These challenges can also lead to inconsistencies in implementation and hinder stakeholder alignment, further complicating problem-solving efforts. While more work is needed to develop definitions and frameworks to guide TDR, to foster alignment and enhance collaborative approaches, it is equally important to avoid imposing rigid or overly standardised approaches. Instead, efforts should focus on fostering critical thinking about TDR's principles and applications, allowing it to remain flexible and adaptable to the specific contexts, needs, and realities it seeks to address. This balance between alignment and flexibility is crucial for maximising TDR's potential in tackling complex challenges.

The review featured the role of TDR and participatory approaches in addressing NTDs predominantly in low- and middle-income countries by emphasising the importance of tailoring interventions to specific socio-political and environmental contexts. These approaches foster partnerships among diverse stakeholders to address the complexity of NTD-related challenges. For instance, the World Health Organization [66] emphasises the importance of integrating human, animal, and environmental health perspectives through a One Health approach, which recognises the interconnectedness of these domains as critical for addressing zoonotic diseases and NTDs. Context-specific approaches such as these ensure that solutions are not only scientifically robust but also culturally appropriate, practical, and relevant to the affected communities, aligning with the immediate and long-term needs of stakeholders.

The findings from this review emphasised the complementary strengths of TDR and participatory approaches in addressing real-world societal challenges through collaboration, despite differences in focus and emphasis. The participatory approaches are particularly effective in fostering trust, cultural relevance, and community ownership by prioritising local knowledge and co-creating solutions with affected populations. In contrast, TDR integrates diverse disciplinary and societal knowledge to address complex systemic issues, offering a broader framework for addressing multifaceted challenges. Together, these approaches create impactful interventions for NTDs, with their success relying on flexibility and adaptability to evolving contexts. Bardosh et al. [67] demonstrated that effective community-based interventions require participatory methods that build on local knowledge and priorities while addressing behavioural and structural constraints. Similarly, Belcher et al. [68] showed that the projects they reviewed effectively combined TDR and participatory approaches, using stakeholder engagement to address real-world problems and influence policies across sectors. These findings emphasise the importance of local engagement and stakeholder empowerment in achieving sustainable outcomes and driving systemic change. However, maximising these strengths effectively requires addressing inherent challenges, including power dynamics and resource constraints.

Stakeholders play a critical yet challenging role in TDR and participatory approaches when applied to NTD studies. Facilitating equitable partnerships requires actively addressing power dynamics, which stem from diverse perspectives, priorities, and levels of influence. Without examining power in relation to social, political, and historical factors, hidden inequalities can reinforce imbalances, marginalise certain groups, and silence dissenting voices. Egid et al. [69] highlight that unaddressed power issues can perpetuate inequities within partnerships. Therefore, acknowledging and addressing power dynamics is not merely an ethical imperative but also crucial for producing research that is both relevant and actionable. [70,71] Tools like participatory mapping, ranking exercises, and collaborative training can help foster equitable collaboration and sustainable, context-specific interventions.

The review identified a lack of reporting on monitoring and evaluation (M&E) frameworks in some of the 17 studies analysed. This gap demonstrates the need for further investigation into whether such frameworks are being adequately implemented or reported. Robust yet flexible M&E frameworks are essential for TDR and participatory approaches addressing NTDs. These frameworks should provide clear guidance for planning, implementation, and evaluation while remaining adaptable to diverse contexts. Cyclical opportunities for reflection and action ensure that these interventions remain responsive to stakeholder needs. Equitable stakeholder engagement and iterative evaluation processes foster inclusivity and cultural sensitivity, driving continuous improvements. The findings of Castro-Diaz et al. [72] highlight the potential of TDR and participatory approaches, when supported by appropriate frameworks, to overcome systemic challenges and create tailored, sustainable interventions. Hence,

developing frameworks that balance robustness with flexibility is crucial for ensuring scientifi-cally sound, contextually relevant, and inclusive solutions that meet stakeholder needs.

Resource constraints, such as limited time and funding, are common challenges in TDR and participatory approaches addressing NTDs. While these approaches are inherently resource-intensive, their combined focus on contextual analysis and stakeholder collabora-tion ensures that interventions are scientifically robust and aligned with local needs, fostering systemic change and sustainability. Policymakers should prioritise creating enabling environ-ments that support these approaches, providing resources and institutional backing to scale context-specific solutions effectively. TDR and participatory approaches serve as a bridge between research and practice by engaging policymakers, practitioners, and communities to co-produce sustainable solutions. Their emphasis on integrating multiple perspectives and focusing on community engagement creates a powerful synergy for tackling systemic issues associated with NTDs.

## Strengths and limitations

This scoping review synthesises diverse studies from a comprehensive database search, identi-fying practical insights and examining the strengths and challenges of TDR and participatory approaches in addressing problems related to NTDs. It advances theoretical understanding while providing a foundation for actionable strategies, making it a valuable resource for both academic and practical applications.

Only peer-reviewed articles were included in this review, while resource constraints excluded unpublished research, non-English articles and manual searches. Unlike systematic reviews, no formal quality assessment was performed. Additionally, the focus on "transdisci-plinary" and related terms may have excluded relevant studies that did not explicitly mention these terms. Another limitation is the absence of a universal definition, methodology, and terminology for TDR [64], which can lead to fragmentation and inconsistency, making it challenging for researchers to integrate their expertise, compare findings and ensure reliable, reproducible results.

## Future directions and study implications

TDR and exploratory approaches are valuable for researchers investigating complex societal issues or sustainability-related topics [37,73], as they enhance the utilisation of research in practice and policy. To encourage broader adoption of these approaches in NTD research, it would be beneficial to provide guidance and training for researchers considering the unique challenges of the NTD field and share reflections and learnings across research teams. [74] Conducting training programmes, reflective workshops, and seminars can equip research-ers with the skills needed to implement these approaches effectively. Collaborative efforts to develop a standardised framework and guidelines for TDR approaches, robust monitoring and evaluation mechanisms and equitable stakeholder engagement are essential to ensure impactful interventions.

## Conclusion

This review highlights the transformative potential of TDR and participatory approaches in addressing the multifaceted challenges of NTDs. These approaches demonstrate complemen-tary strengths, with TDR emphasising interdisciplinary collaboration to address systemic issues and participatory approaches fostering community engagement, trust, and cultural relevance. By bridging disciplines and fostering co-learning, these approaches create innova-tive solutions and contextually relevant interventions that resonate with community needs.

Flexibility and adaptability emerged as crucial for success, as illustrated in studies where participatory methods empowered local stakeholders to actively shape interventions. Addressing power dynamics is essential for equitable stakeholder engagement, to ensure collaboration and ownership of outcomes. Despite their potential, the lack of standard definitions and frameworks for TDR presents challenges, leading to inconsistencies and fragmented approaches across studies. Contextual analysis is vital for tailoring interventions to socio-political and environmental realities. These approaches can strengthen health systems, support sustainable interventions, and reduce health disparities. TDR and participatory approaches also generate data that inform policies and prioritise resources based on community needs. However, challenges such as the lack of robust monitoring mechanisms remain and must be addressed to maximise impact. By embracing TDR and participatory approaches as a central framework, stakeholders can strengthen efforts to combat NTDs, fostering a healthier and more equitable world where communities thrive with improved health and well-being.

The top four lessons from this review aligned with the aim of exploring how TDR and participatory approaches are utilised to address NTD-related challenges:

1. The potential of TDR and participatory approaches in addressing NTD challenges—the review showed that the two approaches were effective through interdisciplinary collaboration and community involvement.

2. Importance of tailoring interventions to context—understanding local socio-political, cultural and environmental situations

3. Lack of standardisation of definitions – need clearer guidelines to improve their implementation

4. Role of flexibility and inclusive engagement—to mitigate power imbalances, ensuring equitable stakeholder engagement

## Supporting information

**S1 Appendix. Working definitions of standard terms used in the study.**
(DOCX)

**S2 Appendix. Search strategy per database.**
(DOCX)

**S3 Appendix. Included studies in the review.**
(DOCX)

**S4 Appendix. A template data extraction instrument.**
(DOCX)

**S5 Appendix. Characteristics of the included studies.**
(DOCX)

**S6 Appendix. Scoping Reviews (Prisma-ScR) Checklist.**
(DOCX)

## Author contributions

**Conceptualization:** Norana Abdul Rahman, Vaikunthan Rajaratnam, Marjolein B. M. Zweekhorst, Ruth M. H. Peters.

**Data curation:** Norana Abdul Rahman, Vaikunthan Rajaratnam, George L. Burchell.

**Formal analysis:** Norana Abdul Rahman, Vaikunthan Rajaratnam.

**Investigation:** Norana Abdul Rahman.

**Methodology:** Norana Abdul Rahman.

**Software:** George L. Burchell.

**Supervision:** Karen Morgan, Mohamed Rusli Abdullah, Marjolein B. M. Zweekhorst, Ruth M. H. Peters.

**Validation:** Norana Abdul Rahman, Vaikunthan Rajaratnam, George L. Burchell.

**Visualization:** Norana Abdul Rahman, Vaikunthan Rajaratnam.

**Writing – original draft:** Norana Abdul Rahman.

**Writing – review & editing:** Norana Abdul Rahman, Vaikunthan Rajaratnam, George L. Burchell, Karen Morgan, Mohamed Rusli Abdullah, Marjolein B. M. Zweekhorst, Ruth M. H. Peters.

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
