## [Decision Letter · Decision Letter 0]

3 Mar 2025

Dear Dr Abdul Rahman,

We are pleased to inform you that your manuscript 'The transdisciplinary research process and participatory research approaches used in the field of neglected tropical diseases : A scoping review' has been provisionally accepted for publication in PLOS Neglected Tropical Diseases.

Best regards,

Uwem Friday Ekpo, PhD

Academic Editor

Qu Cheng

Section Editor

Shaden Kamhawi

co-Editor-in-Chief

Paul Brindley

co-Editor-in-Chief

Reviewer's Responses to Questions

**Key Review Criteria Required for Acceptance?**

**Methods**

-Are the objectives of the study clearly articulated with a clear testable hypothesis stated?

-Is the study design appropriate to address the stated objectives?

-Is the population clearly described and appropriate for the hypothesis being tested?

-Is the sample size sufficient to ensure adequate power to address the hypothesis being tested?

-Were correct statistical analysis used to support conclusions?

-Are there concerns about ethical or regulatory requirements being met?

Reviewer #1: The manuscript "The transdisciplinary research process and participatory research approaches used in

the field of neglected tropical diseases : A scoping review" is a scoping review with clearly stated objectives. The author used a standard referenced methodology for their review which is appropriate for for the study, the authors clearly stated the limitations of the study which is appropriate. The stepwise process were clearly outlined by the authors

Reviewer #2: The objectives of the study are clearly articulated. The study design is appropriate to address the objectives. The sources of data are clearly described. The data analysis is thorough and adequate for the data presented. There are no ethical concerns.

**Results**

-Does the analysis presented match the analysis plan?

-Are the results clearly and completely presented?

-Are the figures (Tables, Images) of sufficient quality for clarity?

Reviewer #1: Yes, the analysis presented match the analysis plan and the results were clearly presented and complete. Results were presented for all the objectives outlines. The tables and figures are well presented and well labelled

Reviewer #2: The analysis presented matches the analysis plan. The results are clearly and compretely presented. The figures are clear and of sufficient clarity to the reader.

**Conclusions**

-Are the conclusions supported by the data presented?

-Are the limitations of analysis clearly described?

-Do the authors discuss how these data can be helpful to advance our understanding of the topic under study?

-Is public health relevance addressed?

Reviewer #1: The conclusions were well supported by the data presented and the limitations of analysis clearly described and which is that they did not include grey literatures. The authors discuss in details how their findings can be helpful to advance our understanding of transdisciplinary research process and participatory research approaches used in

the field of neglected tropical diseases. This topic is of great public health relevance especially because it is becoming cotemporary and trending methods for addressing complex issues regarding the control and elimination of NTDs

Reviewer #2: The conclusions are supported by the data presented. The limitations of the study were clearly described. The authors discuss how the data can be helpful to advance our understanding of the relevance of Transdisciplinary research and Participatory research in the field of NTDs. Public health relevance is addressed.

**Editorial and Data Presentation Modifications?**

Reviewer #1: None

Reviewer #2: No suggestions.

**Summary and General Comments**

Reviewer #1: The manuscript "The transdisciplinary research process and participatory research approaches used in

the field of neglected tropical diseases : A scoping review. It an important work, as the use of transdisciplinary research process and participatory research approaches in the field of neglected tropical diseases is becoming a trending and most effective approach. The authors were able to discuss the importance, benefit of using this research process and some challenges of using the research process. I think the manuscript will be great interest to people who are interested in understanding the benefit of these approach.

Reviewer #2: The study addresses a growing approach to the study of Neglected Tropical Diseases, namely transdisciplinary research process and participatory research. Given their apparent suitability to address the complexity of NTDs the scoping review of studies employing such approaches is welcomed. The study design is appropriate, the methods are adequate and the discussions and conclusions are thorough. The text is clear and concise. There are no ethical concerns.

PLOS authors have the option to publish the peer review history of their article (what does this mean? ). If published, this will include your full peer review and any attached files.

**Do you want your identity to be public for this peer review?** For information about this choice, including consent withdrawal, please see our Privacy Policy .

Reviewer #1: **Yes: ** AKINOLA STEPHEN OLUWOLE

Reviewer #2: **Yes: ** Zoica Bakirtzief

---

## [Editor Report · Acceptance letter]

Dear Dr Abdul Rahman,

We are delighted to inform you that your manuscript, "The transdisciplinary research process and participatory research approachesused in the field of neglected tropical diseases: A scoping review," has been formally accepted for publication in PLOS Neglected Tropical Diseases.

Best regards,

Shaden Kamhawi

co-Editor-in-Chief

Paul Brindley

co-Editor-in-Chief
